# Intracranial In Situ Thermosensitive Hydrogel Delivery of Temozolomide Accomplished by PLGA–PEG–PLGA Triblock Copolymer Blending for GBM Treatment

**DOI:** 10.3390/polym14163368

**Published:** 2022-08-18

**Authors:** Weinan Gu, Ranran Fan, Jingnan Quan, Yi Cheng, Shanshan Wang, Hui Zhang, Aiping Zheng, Shenghan Song

**Affiliations:** 1School of Pharmacy, Xuzhou Medical University, Xuzhou 221004, China; 2State Key Laboratory of Toxicology and Medical Countermeasures, Institute of Pharmacology and Toxicology, Beijing 100850, China; 3Department of Vascular Surgery, Beijing Chaoyang Hospital, Capital Medical University, Beijing 100020, China

**Keywords:** glioblastoma relapse, interstitial chemotherapy, thermosensitive, PLGA–PEG–PLGA, triblock copolymer, polymer blending, temozolomide

## Abstract

Glioblastoma (GBM) recurrence after surgical excision has grown to be a formidable obstacle to conquer. In this research, biodegradable thermosensitive triblock copolymer, poly(D, L–lactic acid–co–glycolic acid)–b–poly(ethylene glycol)–b–poly(D, L–lactic acid–co–glycolic acid (PLGA–PEG–PLGA) was utilized as the drug delivery system, loading with micronized temozolomide(micro-TMZ) to form an in situ drug–gel depot inside the resection cavity. The rheology studies revealed the viscoelastic profile of hydrogel under various conditions. To examine the molecular characteristics that affect gelation temperature, ^1^H–NMR, inverse gated decoupling ^13^C–NMR, and GPC were utilized. Cryo-SEM and XRD were intended to disclose the appearance of the hydrogel and the micro-TMZ existence state. We worked out how to blend polymers to modify the gelation point (T_gel_) and fit the correlation between T_gel_ and other dependent variables using linear regression. To simulate hydrogel dissolution in cerebrospinal fluid, a membraneless dissolution approach was used. In vitro, micro-TMZ@PLGA–PEG–PLGA hydrogel exhibited Korsmeyer–Peppas and zero–order release kinetics in response to varying drug loading, and in vivo, it suppressed GBM recurrence at an astoundingly high rate. Micro-TMZ@PLGA–PEG–PLGA demonstrates a safer and more effective form of chemotherapy than intraperitoneal TMZ injection, resulting in a spectacular survival rate (40%, n = 10) that is much more than intraperitoneal TMZ injection (22%, n = 9). By proving the viability and efficacy of micro-TMZ@PLGA–PEG–PLGA hydrogel, our research established a novel chemotherapeutic strategy for treating GBM recurrence.

## 1. Introduction

The stage IV astroglioma tumor known as glioblastoma multiforme is the most aggressive and has the poorest prognosis, with the majority of patients dying from tumor relapse after surgical resection [1]. Only 5% of GBM patients survive for five years [2,3]. To avoid tumor relapses following surgery, chemotherapy and radiotherapy are frequently used [4]. In an effort to improve the efficacy of chemotherapy, the FDA has approved the Gliadel^®^ wafer, which contains BCNU and a biodegradable polifeprosan carrier material [5,6] and is implanted into GBM patients’ resection cavities. However, the average prolonged survival time only increased by 2 months, which was disappointing, and occasionally adverse events happened [7]. TMZ was cited as the first–line drug in the majority of GBM clinical treatment recommendations from around the world [8]. Despite having a marginally higher ability than conventional chemotherapeutic medications to cross the blood–brain barrier (BBB), the number of agents that ultimately targeted the site of the brain tumor was still rather small due to their instability in bodily fluids and poor lipophilicity [9]. Numerous studies have demonstrated the effectiveness of using intracranial preparation to create microspheres and nanoparticles to address this problem [10,11].

To enable various drug delivery strategies, from proteins such as exenatide [12], lixisenatide [13], dexamethasone [14], and probes [15] to chemotherapeutic medicines such as doxorubicin [16], gemcitabine [17], PLGA–PEG–PLGA triblock copolymer was used as the carrier material in a thermosensitive hydrogel form [18]. The traditional PLGA–PEG–PLGA could also be modified via polymerizing an iodinated trimethylene carbonate monomer into the two ends of PEG to reveal the degradation in deep tissue [19]. Depending on the molecular weight of the PEG unit and PLGA unit, it can be used as a water–insoluble nanoparticle carrier [20], a thermosensitive hydrogel (hydrophilic–hydrophobic balanced), or a polymer solution [21] (completely hydrophilic). Due to its amphiphilic nature, it would naturally form a PLGA-core–PEG-shell micelle in a surrounding aquatic environment. When the temperature rises, PEG segment would become more hydrophobic, and micelles would gather together and entangle. According to Wu et al., the gelation is mainly triggered by dehydration of PEG [22], the bridge that PEG segment builds also contributes to the formation of hydrogels [23].

Although the exact feeding weight ratio of PEG, LA and GA can be designed for each segment’s M_w_ before synthesizing, the outcome may deviate somewhat from theory. Raw PEG1500 is also a low–molecular–weight polymer with a variety of polydispersity indexes; a few block differences would upset the equilibrium between the hydrophilicity and hydrophobicity of the PEG chain and the PLGA chain. Therefore, gelation qualities may not be as expected and may not be apparent until it becomes entirely water dissolved. According to Yu [24], a PLGA–PEG–PLGA solution was blended with an insoluble PLGA–PEG–PLGA precipitate until it becomes completely dissolved in the polymer solution to create the desired thermogelling characteristic. This finding suggests that it is possible to alter the thermosensitive features of PLGA–PEG–PLGA by polymer blending.

To protect brain function during surgery, patients’ bodies are often kept at an appropriate low temperature (moderate hypothermia) to reduce brain oxygen use [25]. The appropriate gelation temperature should thus be flexible. The hydrogel is a liquid with high flowability at room temperature that, due to its temperature sensitivity, turns into a gel at around 32 °C and maintains the gel state at body temperature of 37 °C [26,27]. Liang et al. used a thermosensitive peptide loaded with interferon for local application in the resection cavity of gliomas, making it possible to inhibit recurrence with local application of a temperature–sensitive gel after glioma surgery. However, according to the authors’ description, better efficacy was achieved only when it was spatiotemporally combined with TMZ chemotherapy [28]. Therefore, TMZ plays a crucial role in the process of GBM recurrence inhibition. It is also confirmed that protein expression of CXCR4, MMP2, MMP9, and VEGF can be decreased under either high dosage or long exposure of TMZ, which are crucial cytokines in GBM relapse [29]. Combined with the available literature, if TMZ could be applied locally in the form of a drug–loaded gel in the resection cavity of GBM, its slow–release, long–lasting effect would continue to act on the remaining GBM cells to achieve complete inhibition to the point of cure.

Therefore, the present work considers TMZ in combination with a thermosensitive hydrogel for topical application in the GBM resection cavity to inhibit recurrence. The formability, safety, efficacy, and stability of the PLGA–PEG–PLGA temperature–sensitive hydrogel were proven in this work. TMZ can slowly release from the hydrogel due to the in situ drug depot, resulting in high local concentrations and penetrating into residual brain tumor tissue. This dosage form offers a low dose and reaches therapeutic goals with only one intraoperative administration, considerably improving patient compliance as compared to traditional TMZ capsules and injections. By using hydrogel slow release [30,31,32], it is possible to reach the highest local TMZ concentrations while lowering systemic toxicities [33]. The formulation offers a novel therapeutic alternative for individuals with astrocytic glioma [34,35] and has great biocompatibility while displaying effective prevention of GBM recurrence [36].

## 2. Materials and Methods

### 2.1. Materials

Temozolomide (TMZ), 99.8% pure, Lot No. 04030155210204 was purchased from Hairun Pharmaceutical Co., Ltd., Nanjing, China. PLGA–PEG–PLGA triblock copolymer, Lot No. 21062206 (theoretical M_w_: 2000–1500–2000, Copolymer–1); Lot No. 21113003(theoretical M_w_: 1700–1500–1700, Copolymer–2); Lot No. 21090902(theoretical M_w_: 1700–1500–1700, Copolymer–3); Lot No. 21120701(theoretical M_w_: 1700–1500–1700, Copolymer–4) were obtained from Academy of Biomedical Sciences Co., Ltd., Jinan, China. PLGA–PEG–PLGA triblock copolymer (theoretical M_w_: 1750–1500–1750, Copolymer–5) was purchased from Jinan Daigang Bioengineering Co., Ltd., Jinan, China. U87–MG/Luc was purchased from Fuheng Biotechnology Co., Ltd., Shanghai, China. D–Luciferin (≥98%) was purchased from Yuanye Bio–Technology Co., Ltd., Shanghai, China. MTT Assay kit was bought from Edison Biotechnology Co., Ltd., Yancheng, China. Sodium chloride; Potassium dihydrogen phosphate; Potassium chloride; Dimethyl sulfoxide and Glacial acetic acid were bought from Sinopharm Chemical Reagent Co., Ltd., Shanghai, China. Methanol (HPLC grade) was obtained from Thermo Fisher Scientific (China) Co., Ltd., Shanghai, China. Sodium hexane sulfonate was purchased from Yuanye Bio–Technology Co., Ltd., Shanghai, China. All water used was double distilled water purchased from Watsons, Beijing, China.

### 2.2. Hydrogel Preparation

The concentration of PLGA–PEG–PLGA solution is denoted as a weight percentage (wt%), which is equal to the mass of solute divided by the mass of solution. Due to its stickiness, which manifests as a sticky paste at room temperature, the material cannot be accurately picked up as intended. By adding a given amount of water, the concentration can be altered flexibly.

To comprehend the gelation behavior of several PLGA–PEG–PLGA materials, gel solutions of varying concentrations (5; 10; 15; 20; 25; 30 wt%) are created. To remove insoluble particles, a filter membrane with a 0.45 μm microporous is required. Gelation and precipitation temperature were evaluated using the Tube–Reverting Method. Increasing the water bath temperature by 1 °C per min and inverting the tube at each temperature. The sol–gel point is reached when the solution stops flowing after 30 s of inversion. Consequently, the gelation point is denoted as T_gel_ [37,38]. The precipitation point should be recorded when reaching a new temperature level. This is the sol–gel transition point, as well as the point at which T_precipitation_ occurs. Generally, the system would become opaque, and there would be a large number of white clusters, which makes the gel easy to flow again.

However, the given gelation features of the PLGA–PEG–PLGA thermosensitive hydrogel may be unexpected, such as the material’s abnormally low gelation temperature, which causes it to form a gel at room temperature (around 25 °C) in a matter of seconds, or its inability to form a gel at all. These are unsuitable for the application of a thermosensitive PLGA–PEG–PLGA hydrogel to patients. We hypothesized a strategy to modify the gelation point by polymer mixing and fortunately, demonstrated its applicability successfully, as evidenced by the literature [24,39,40]. The fundamental concept is to combine the high T_gel_ and low T_gel_ PLGA–PEG–PLGA gel solutions at the same concentration but in various volume ratios to produce a new solution that maintains the newly generated T_gel_ within the interval of the T_gel_s. Here, we combined Copolymer–1 and Copolymer–2 in various 20 wt% ratios to produce T_gel_ at roughly 32 °C, which is liquid at room temperature and becomes a gel at body temperature.

TMZ is a drug with crystallinity ranging from 30 to 60 μm in drug size. In order to prevent rapid sedimentation after mixing with the hydrogel but before application, the particle size should be reduced in accordance with Stokes’ Law. Utilizing a superfine powder airflow mill, the particle size of TMZ was efficiently reduced.

### 2.3. Physicochemical Properties of PLGA–PEG–PLGA and Hydrogels

We used cryo-SEM to determine the microstructures of the PLGA–PEG–PLGA hydrogel and micro-TMZ dispersion in the hydrogel in addition to NMR spectra and GPC to determine the molecular characteristics. The XRD test was used to evaluate the interaction and crystallinity of micro-TMZ and PLGA–PEG–PLGA hydrogels. Experiments including rheological oscillation tests were conducted to evaluate the macroscopic viscoelasticity of the PLGA–PEG–PLGA hydrogel, such as T_gel_ and modulus, under various conditions. To confirm that this strategy of medication administration is viable and has the potential to surpass current methods, the membraneless dissolving method was used to analyze drug release.

#### 2.3.1. NMR Spectrum

The PLGA–PEG–PLGA gel solution must be lyophilized to entirely eliminate any leftover water. In order to obtain a transparent solution, approximately 50 mg of PLGA–PEG–PLGA was dissolved in 0.5 mL of deuterated chloroform (CDCl_3_) and then placed in an NMR tube. The experiment utilized 600 MHz AVANCE NEO NMR equipment, Bruker Scientific Technology (Beijing) Co., Ltd., Beijing, China. The mode of detection was inverse gated decoupled (IGD) ^13^C–NMR, which implies that the decoupler is turned off during the relaxation delay period and turned on during the sampling period. The pulse sequence was set to “zgig” mode in order to eradicate the NOE effect of carbon so that the amount of carbons of specific chemical shift is proportional to the area under curve and the carbon quantification is enabled [41,42]. A total of 8 and 512 times were allotted for ^1^H–NMR and ^13^C–NMR scanning, respectively. The data processing application utilized was TopSpin, version 4.0.9 by Bruker Co., Billerica, MA, USA.

#### 2.3.2. GPC

In a volumetric flask, a PLGA–PEG–PLGA tetrahydrofuran solution containing 1 mg/mL was prepared. The substance was analyzed using gel permeation chromatography (GPC) using a refractive index detector (RID) (SHRI–10, Shenghan Chromatograph Technology Co., Ltd., Qingdao, China) on a Thermo U3000 gel permeation chromatography(Thermo Fisher Scientific (China) Co., Ltd., Shanghai, China). To assess molecular weights ranging from 50 to 2,000,000, one Waters styragel Guard pre-column and three Waters styragel columns were utilized (HR4, HR4E, HR3). The volume of the sample was 100 μL, the temperature of the column and the RID was 40 °C, and the flow rate of the tetrahydrofuran mobile phase was 1 mL/min.

#### 2.3.3. Visual Inspection

The visual examination was performed in front of a dark background to obtain a clear macro-image of the status of the thermosensitive PLGA–PEG–PLGA solution at different temperatures in a photo booth.

#### 2.3.4. Microstructure of Hydrogel

As with the majority of hydrogels [43,44], the PLGA–PEG–PLGA thermosensitive hydrogel’s physical network structure is connected by temperature–sensitive micelles. After the PLGA–PEG–PLGA solution had formed a gel at 37 °C on a heating panel [45], the hydrogel was instantly frozen in liquid nitrogen at −196 °C and freeze–fractured to obtain a fine cross–sectional image in a Frozen Transfer Preparation System by QUORUM Co., Ltd., East Sussex, UK. The sublimation was then treated at −70 °C for 30 min to eliminate as much surface water as possible without compromising the integrity of the structure. In order to attain a high resolution, samples were sputtered with a 10 nm layer of gold plasma. The internal structure of the PLGA–PEG–PLGA hydrogel was observed using a JSM–7900F cryo-SEM, JEOL (Beijing) Co., Ltd., Beijing, China and Graphical User Interface Software by JEOL, Co., Ltd., Tokyo, Japan.

#### 2.3.5. X-ray Diffraction Analysis

Crystallinity of TMZ, micro-TMZ, PLGA–PEG–PLGA and micro-TMZ@PLGA–PEG–PLGA was measured using Powder X-ray Diffraction (XRD). With a 2 °C per min scanning speed, samples were all patterned at the 2–theta from 5° to 90°. The tube voltage was 40 kV and tube current was 40 mA. The excitation of the Cu target resulted in a lambda value of 1.5418 Å.

#### 2.3.6. Particle Size Distribution

RODOS Laser Particle Size Detector (Sympatec Gmbh, Suzhou, China) was used to examine the particle size of TMZ and micro-TMZ in a dry state without any media. The dispersion pressure generated by the vacuum module was set to 1 bar for complete scattering of the powder. The trigger limit was set at 5% in that about 10 mg of the sample is sufficient for detection. The software used for the instrument control and visualization is PAQXOS, version 5.0.1 by Sympatec Gmbh, Wolfenbüttel, Germany.

#### 2.3.7. Rheological Tests

The most significant and practical method of learning about a semi-viscoelastic solid’s qualities is through rheological experiments. To evaluate hydrogels under various programs, TA Instruments HR10 Discovery Rotation Dynamic Rheometer (Waters Technologies (Shanghai) Ltd., Co., Shanghai, China) was used [46]. The program we decided on was the oscillating temperature ramp program, which increased the temperature by 2 °C/min from 20 to 50 °C. The frequency was 1 Hz, and the strain was 1%. We selected a 20 mm parallel–plate solvent trap on a Peltier surface as our geometry. A 500 μm gap was established, and the testing volume of the hydrogel was approximately 0.18 mL.

#### 2.3.8. In Vitro Drug Release

To the greatest extent possible, the membraneless dissolving approach was used to mimic the gel dissolution process in the interstitial brain, as documented by the literature [47,48,49]. The dissolution vessel consisted of 10 mL screw–top bottles. A stable micro-TMZ@PLGA–PEG–PLGA hydrogel was created by weighing various amounts of micro-TMZ into a flask, mixing them with 1 mL of PLGA–PEG–PLGA hydrogel (n = 3), and then heating the mixture for 10 min at 37 °C. The hydrogel dissolution medium was prepared in advance and contained sodium chloride, potassium dihydrogen phosphate, and potassium chloride. The pH of the medium was close to 7.4, and the osmotic pressure was close to 290 mOsm/L to simulate artificial cerebrospinal fluid. Then 4 mL of medium was drawn from each point for HPLC detection after filtering through a 0.45 μm Nylon 66 microporous filter membrane after adding 5 mL of medium on top of the gel [50]. Gels were placed on a 37 °C shaking bed at 50 rpm/min. Next, 4 mL of fresh medium was gently sucked out on top for the next detection point. The detection points were 1, 2, 6, 12, 24, 48, 72, 96, 120 etc., every 24 h afterwards.

With a flow rate of 1 mL/min and a mobile phase made up of 96% acetic water solution (0.5 *v*/*v* %) and 4% methanol containing 0.005 mol/L sodium hexane sulfonate, HPLC was used to measure the sample, and 10 μL was the injection volume. The detection period was 29 min at a column temperature of 30 °C for the reverse–phase OSAKA SODA MGII C_18_ chromatography column, 4.6 mm × 250 mm. Agilent ChemStation (version 12.212, Santa Clara, CA, USA) was used as the software for HPLC data acquisition and processing.

The release data were examined using traditional models. Drug release parameters were calculated by several mathematical models: zero–order (1), first–order (2), Higuchi (3) and Korsmeyer–Peppas model (4). Release data were fitted into the model equations in order to delineate a drug release mechanism from the formulations [51,52].
(1)$$Q_{t}=Q_{0}+K_{0}\times t$$
(2)$$logQ_{t}=logQ_{0}+K_{1}\times t/2.303$$
(3)$$Q_{t}=K_{h}\times t^{\frac{1}{2}}$$

Peppas and Korsmeyer, in their published work, proposed a more complex semi-empirical power relation.
(4)$$Q_{t}/Q_{\infty }=K_{p}\times t^{n}$$

*Q_t_*, *Q_0_*, and *Q_∞_* denote, respectively, the cumulative quantity of drug released at time *t*, the beginning amount of drug, and the total amount of drug in a dosage form. *K_0_* is the constant of zero–order release rate calculated by plotting *Q_t_* versus *t*, which is also a constant related to diffusivity as well as structural and geometric features. The release rate constant *K_1_* is derived by plotting log(*Q_t_*/*Q_1_*) against time. *K_h_* is the Higuchi release rate constant calculated by plotting *Q_t_* versus the square root of time. *K_p_* signifies the release rate constant of the Korsmeyer–Peppas model, whereas *n* denotes the release exponent that distinguishes various release processes. Calculate the parameters *K_p_* and n by plotting log(*Q_t_*/*Q_1_*) against the logarithm of *t*.

Equation (4) is simplified to Equation (3) when the release exponent *n* = 0.5 for a slab geometry, which is referred to as case I or Fickian diffusion transport, whereas 0.5 < *n* ≤ 1.0 refers to case II or swelling–controlled transport and intermediate/anomalous transport, and *n* > 1 refers to non-Fickian diffusion.

#### 2.3.9. Thermo–Reversibility Research

The reversibility research on PLGA–PEG–PLGA hydrogel was conducted. There was not enough data to assess if the hydrogel would recover from heat precipitation or a frozen state [53]. In order to ascertain if heat or freeze–thaw will alter the gelation properties, it was necessary to measure the T_gel_ and the modulus (G′, G″) of the PLGA–PEG–PLGA hydrogel recovery under high temperatures. Three programs for heat recovery were set up: from 4 to 50 °C to simulate the heat–precipitation process and to measure the T_gel_ at the same time, from 50 to 4 °C to simulate placing the gel in a cold environment generally. Then, the gel was kept under 4 °C for as long as possible to test whether the precipitated gel could recover its original T_gel_. This is one whole circulation. In order to measure the T_gel_ without inducing precipitation, the freeze–thaw recovery process was carried out in four steps: from 4 to 37 °C and then from 37 to 4 °C. In order to simulate the frozen condition, the program was set to −20 °C with a 300 s oscillation duration. In order to depict the thawing process, the environment was set to 4 °C with an oscillation time of 300 s. With a 1 Hz frequency and a 1% strain, the entire temperature ramp rate was set at 2 °C/min. All samples were freshly blended before rheology tests in a specific volume ratio of Copolymer–1 with Copolymer–2.

#### 2.3.10. Statistical Analysis

IBM SPSS Statistics (version 25, IBM Corp., Armonk, NY, USA) was utilized as the main software for statistical analysis on MacOS 12.2.

### 2.4. Cell and Animal Experiments

#### 2.4.1. Cell Culture

The human–origin malignant glioblastoma cell U87–MG/Luc was cultivated for the subsequent cell and animal experiments. The cell was lentivirus transfected to add a luciferase label for later IVIS detection. It was raised in a 37 °C, 5% CO_2_ Cell Culture Chamber using Dulbecco’s Modified Eagle Medium (DMEM) containing 10% Fetal Bovine Serum and 1% penicillin–streptomycin solution.

#### 2.4.2. In Vitro Hydrogel Cytotoxicity Assay

As is reported that high dosage hydrogel during cell viability test would suffocate the cells, which would lead to false positive results [54], the concentration of PLGA–PEG–PLGA was set to an optimum concentration that would not form a gel in the 96–well plate. The test wells were filled with PLGA–PEG–PLGA in DMEM (containing 100 U/mL penicillin, 100 mg/mL streptomycin, and 10% fetal bovine serum) at successively lower doses (10, 7.5, 5, 2.5, 1, 0.5, 0.1, 0.05 mg/mL). Due to its volatility, different TMZ concentrations in DMEM (2, 1.5, 1, 0.5, 0.2, 0.1, 0.05, and 0.01 mg/mL) were formulated precisely before being applied to the wells (2, 1.5, 1, 0.5, 0.2, 0.1, 0.05, and 0.01 mg/mL). It was also revealed how the combination of TMZ and PLGA–PEG–PLGA hydrogel causes cytotoxicity. The concentration of PLGA–PEG–PLGA was set at 1 mg/mL, whereas the concentration of TMZ was the same as in the TMZ alone group. After 48 h of 37 °C incubation in a humidified environment containing 5% CO_2_, 10 μL of MTT solution was added to each well. After three more hours of maintenance, 10 μL of DMSO was added to dissolve the formazan. The vitality of the cells on the plate was quantified using an ELISA reader. The wavelength of detection was 570 nm. The software Prism, version 9.0 (Graphpad Software LLC, San Diego, CA, USA) was used to calculate IC_50_ values [55].
Cell Viability = (OD_Drug − OD_blank)/(OD_control − OD_blank) × 100%

#### 2.4.3. Animal Experiments

All mice used in the research, including balb/c and balb/c nude mice, were 6–week–old females weighing around 20 to 22 g, acquired from Vital River Laboratory Animal Technology Co., Ltd., Beijing, China. All studies were conducted in accordance with the Institutional Animal Care and Use Committee of the National Natural Science Foundation of China (IACUC–DWZX–2020–639). According to the Guide for the Care and Use of Laboratory Animals, we cared for the animals. The Animal Ethics Committee of the Institute of Pharmacology and Toxicology, Beijing China, granted approval for the testing conducted.

#### 2.4.4. Procedure of GBM Modeling in Balb/c Nude Mice

Balb/c nude mice were orthotopically placed in a stereotactic apparatus after being anesthetized with urethan solution at a dose of 1250 mg/kg. After normal scalp disinfection with 75% ethanol and skin preservation, a 5 mm incision was made to expose the parietal bone using ophthalmic scissors. Soft tissue and fascia were removed with hemostatic forceps prior to digging a 0.2 mm diameter hole with a high–velocity cranial drill at the 3 mm posterior coronal suture and 2 mm lateral bregma point. The microsyringe was thoroughly sterilized to remove any microorganisms that could cause meningitis or another devastating infection. The injection volume was set at 8 μL, as mentioned in the literature [28], to assure the success of the modeling and prevent mortality. The U87–MG/Luc cell concentration in DMEM was 1 × 10^6^/mL, and the aspiration volume was 8 μL. Prior to injection, the depth of the needle was fixed at 3 mm. The injection was administered at a rate of 0.5 μL per min, and the entire treatment lasted at least 15 min. Before being placed on a 37 °C temperature–controlling panel, the incision was sutured with a biocompatible PGA thread and re-disinfected. Notably, mice whose weights decreased by roughly 10% after modeling or did not demonstrate chemiluminescence following intraperitoneal injection with 150 mg/kg D–Luciferin was to be euthanized. Mice with equal chemiluminescence sizes were paired together for further treatment.

#### 2.4.5. GBM Resection Surgery Process

The scalp was cleaned with ethanol following urethan anesthesia and stereotactic apparatus fixation. On the last healed wound, a 5 mm incision was made prior to fascia removal. Before removing as many GBM cells as possible with a 1.5 mm Miltex^TM^ Biopsy Punch (Integra LifeSciences Co., Plainsboro, NJ, USA), a high–speed cranial drill was used to enlarge the previously formed bone hole to a diameter of roughly 1.5 to 2 mm. The excision was performed at a 3 mm depth using a 20 s rolling motion in a clockwise direction to separate the GBM tissue. The removal of material was accomplished with disinfected forceps. At this time, in situ hydrogel groups must cast hydrogel with or without a medication load. After re-disinfecting the bone hole, a square piece of sterile hemostatic sponge measuring 2 by 2 mm was applied to aid wound healing.

After urethan anesthesia and stereotactic equipment fixation, ethanol was utilized to clean the scalp. Prior to fascia removal, a 5 mm incision was made on the most recently healed site. Prior to removing as many GBM cells as possible with a 1.5 mm Miltex^TM^ Biopsy Punch, a high–speed cranial drill was used to widen the previously produced bone hole to around 1.5 to 2 mm in diameter. To separate the GBM tissue, the excision was conducted at a 3 mm depth using a 20 s rolling motion in a clockwise orientation. Utilizing disinfected forceps, the substance was extracted. In situ hydrogel groups were currently cast hydrogel with or without a drug load. After re-disinfecting the bone hole, a 2 by 2 mm square of sterile hemostatic sponge was used to promote wound healing.

#### 2.4.6. Biosafety Evaluation

Numerous studies have demonstrated that PLGA–PEG–PLGA possesses exceptional biocompatibility and biodegradability, making it a great material for an in situ drug–hydrogel delivery system [56,57]. In order to compare the safety of micro-TMZ@PLGA–PEG–PLGA hydrogel to that of conventional drug delivery methods, n = 6 balb/c mice were divided into three groups, each receiving an intracranial injection of 8 μL of Normal Saline, micro-TMZ@PLGA–PEG–PLGA hydrogel, and a blank group. For 14 days, body weight was monitored and H&E staining was performed to evaluate apoptosis and inflammation in cells. On day 14, the brains were removed, and H&E staining was conducted according to protocol.

#### 2.4.7. Suppression of GBM Relapse

Balb/c nude mice were used in this experiment to estimate the efficacy of micro-TMZ@PLGA–PEG–PLGA hydrogel intracranially. Modeling was performed on day 0. One week later on day 6, it would have a clear image of chemiluminescence under IVIS (PerkinElmer), demonstrating the proliferation of GBM cells in situ after being given 150 mg/kg D–luciferin intraperitoneally; mice with similar chemiluminescence were grouped. GBM excision surgery was performed on Day 7, following which four groups were established: n = 6 for model mice without treatment (surgery group), n = 6 for model mice with PLGA–PEG–PLGA hydrogel intracranial injection alone (surgery + PLGA–PEG–PLGA hydrogel group), n = 6 for model mice with TMZ solution intraperitoneal injection (surgery + TMZ intraperitoneal group), and n = 6 for model mice with micro-TMZ@PLGA–PEG–PLGA intracranial injection (surgery + micro-TMZ@PLGA–PEG–PLGA hydrogel group). The dosage of TMZ intraperitoneal injection and micro-TMZ@PLGA–PEG–PLGA hydrogel was 50 mg/kg and 100 mg/mL × 8 μL, respectively. IVIS spectrum was observed every 7 days to reveal the size and extent of the GBM tumor and prognosis. Survival time was continuously observed within 60 days. The exposure time of IVIS was set at 1 s, the chemiluminescence range was set between 50 and 1000. SPSS 25.0 was utilized to statistically assess the efficacy of GBM relapse suppression utilizing log–rank analysis and the Kaplan–Meier Survival Curve.

## 3. Results and Discussion

### 3.1. Macro Appearance

Due to the instability of TMZ in a neutral or alkaline aqueous environment, this formulation (micro-TMZ@PLGA–PEG–PLGA thermosensitive hydrogel) is formed of PLGA–PEG–PLGA blank hydrogel and micro-TMZ separately, and is created directly prior to use or to characterize. It is complete once the blank PLGA–PEG–PLGA thermosensitive hydrogel and the micro-TMZ have been mixed uniformly to form a homogenous solution.

When PLGA–PEG–PLGA is completely dissolved in water under a certain temperature, often 4 °C, a transparent solution that glows pale blue is produced (Figure 1B). When the system is in solution, i.e., no gelation is occurring, the Z–mean size of the spherical micelles is about 37.92 nm, which is much lower than the membrane diameter of 450 nm; thus theoretically, there is no drop in content due to interception at all. The rheological experiments also demonstrate that the gelling properties have not changed. This is how PLGA–PEG–PLGA thermosensitive hydrogel is typically created. Due to the PLGA–PEG–PLGA’s high viscosity and excessive stickiness, the dissolving process takes up the majority of the experiment’s duration. Depending on the molecular weight, polydispersity indices, or even residual PEG left over from previous batches of manufacturing or the dissolution temperature, the whole dissolving process might take up to a week to complete.

Classical Tube–Reverting Method to measure the phase transition point, however, is somewhat subjective. As the temperature rises, when it comes to the gelation point, the gel may become less flowable, but the deformation of the hydrogel occurs within the entire 30 s. In the subsequent rheological results, it is shown that this is a period where the loss modulus and the storage modulus show an exponential increase, but at this interval, the storage modulus still does not exceed the loss modulus, and the system still has a liquid nature at this point; only the flow becomes poor and is always mistakenly considered to be gelling. Therefore, the gelation point identified by this method is intuitive and convenient but is not accurate. It is the same as the precipitation point determination. This is merely a rough estimate of the basic properties of this batch of PLGA–PEG–PLGA. Line connecting the sites of gelation and precipitation at different concentrations are drawn, and a classic thermosensitive gelation “C” curve will appear, dividing the coordinate plane into three areas: sol state down below, gel state in the middle, and sol (precipitate) state on top [58,59] (Figure 1A).

### 3.2. Rheology Analysis

The rheology test of thermosensitive hydrogels is to accurately quantify T_gel_ by oscillation temperature ramp programs. G′ and G″ is shown regarding storage modulus and loss modulus, which represents solid properties and liquid properties of semi-fluid. G″ is always greater than G′ when the system is an elastic flowable fluid. When it comes to the first crossover of G′ and G″ of typical thermosensitive hydrogel, it is the sol–gel transition which is defined as the T_gel_ [60,61]. T_precipitation_ is called when G″ is greater than G′ again. At this moment, it represents a fierce gathering of PLGA–PEG–PLGA micelles, which leads the phase to precipitate, dividing the system into two phases (Figure 2A).

As soon as PLGA–PEG–PLGA was synthesized and dissolved in solution, experiments were conducted to determine the phase diagram. However, it perplexed us for quite some time that even though the theoretical molecular properties and synthesis process were identical, the phase diagram was not identical, particularly the gelation point, which was either too high to form a gel at body temperature or so low that it became insoluble after only a few seconds at room temperature. Theoretically, Copolymer–2, Copolymer–3, Copolymer–4 and Copolymer–5 were all 1700–1500–1700, yet their T_gel_ values varied greatly, as well as the insolubility of Copolymer–4. By adjusting the concentrations of PLGA–PEG–PLGA, the gelation temperature cannot be altered considerably, but the modulus can also be changed, hence influencing the hydrogel’s strength (Figure 2B). When we were at a loss, the concept of blending was proposed. Two PLGA–PEG–PLGA hydrogel solutions with differing gelation points were blended, and the fact that PLGA–PEG–PLGA can be customized by blending intrigued us all. Here, blended samples of Copolymer–1 and Copolymer–2 in the ratio of 1 to 4 produced a series of T_gel_ (Figure 1A), which corresponds to the rheology tests (Figure 2A). Thus, it became simple to customize specific T_gel_s as desired. Figure 2C,D depicts the regularity of T_gel_ under varied blending ratios. T_gel_s were created successfully with the modulus minimally altered.

The preparation is not a blank gel; thus, the micro-TMZ@PLGA–PEG–PLGA hydrogel was subjected to a rheology test to determine its flowability and thermosensitivity prior to its application to the GBM excision cavity. The results showed nearly no T_gel_ changes (RSD = 1%), but the G′ tends to marginally increase relative to the blank gel as the micro-TMZ loading is increased (Figure 2E). It is possible that this is related to the decrease in gel fluidity and increase in strength caused by micro-TMZ embedded in the PLGA–PEG–PLGA hydrogel mesh structure.

### 3.3. Microstructure of the Hydrogel and Formulation

From Figure 3A, it was observed that TMZ is a powder with square crystallinity, and the particle size was approximately 40 μm under SEM. It was evident that the micronization process reduced the particle size and eliminated the square crystallinity.

Cryo-SEM also revealed the conventional hydrogel network architecture (Figure 3B). All structures of hydrogel mesh could be investigated. Micelles with a spherical shape can be detected in Copolymer–2. As T_gel_ decreases, spherical micelles dissipate, the micelles become violently entangled, and thick plate micelles develop. Densely dotted plate micelles illustrate the entanglement well under the observation of Cyro-SEM with 10,000 times magnification. The lower the T_gel_, the greater the degree of entanglement at 37 °C, and as the thickness of the plate increases, so does the mesh size of the hydrogel, further demonstrating that temperature is the primary trigger for gelation and micelles play an important role in the property. Additionally, the gelation state is determined by the degree of micelle aggregation. The mesh size of B–1 was around 4 μm, allowing it to encapsulate micro-TMZ in a crystalline form with a variety of morphologies (Figure 3C).

### 3.4. NMR Spectra and GPC Diagrams Revealing Molecular Properties and Statistic Analysis

As follows, ^1^H–NMR and inverse gated decoupling (IGD) ^13^C–NMR spectra were acquired (Figure 4A,B). Evaluation of chemical shifts in the ^1^H–NMR spectrum reveals different types of hydrogen, and the ratio of area integration at various chemical shifts can be used to calculate the number average molecular weight. Literature [62,63,64] defines peaks at δ 1.2–1.6, δ 3.4–3.7, δ 4.1–4.3, δ 4.5–4.9, and δ 5.0–5.3 as hydrogen integration of methyl in LA, methylene in PEG chain (without both ends), methylene at both ends of PEG, methylene in GA, and hypomethyl in LA. Theoretically, based on the integration area under δ 3.4–3.7, we may calculate and quantify integration using the manufacturer–provided molecular weight of the PEG section. However, the molecular weight of PEG segment varies widely from batch to batch that it cannot be classified as PEG1500, despite its theoretical designation as such. The precision of PLGA–PEG–PLGA molecular weight calculations derived from ^1^H–NMR is therefore questionable [65].

IGD ^13^C–NMR was applied to minimize the influence of the Nuclear Overhauser Effect (NOE) gain and relaxation time of ^13^C in diverse chemical environments, allowing for the quantification of ^13^C in PLGA–PEG–PLGA. δ 15–18, δ 59–62, δ 63–65, δ 68–69.5, δ 69.5–72, and δ 166–178 are recognized as the integration of ^13^C peaks, methyl carbon in LA, methylene carbon in GA, both carbon ends of PEG, hypomethyl carbon in LA, methylene carbon in PEG (without both ends), and carbonyl carbon in both LA and GA, which is an sp^2^ hybridization. It would be more precise to determine the number average molecular weight (M_n_) if it were known that the number of carbon atoms at both ends of PEG is two, with a chemical shift of δ 63–65. Once this concept is understood, determining other carbons with distinct chemical changes is simple.

As indicated in Table 1, ^13^C–NMR spectra of PLGA–PEG–PLGA under varied blending ratios have been utilized to calculate molecular properties. Minor alterations in the molecular weight of PLGA or PEG may result in pronounced T_gel_ variety. It can be observed that the ratio of PLGA to PEG (R_PLGA/PEG_) is inversely proportional to T_gel_, which is consistent with the literature [66,67].

After blending, the liquid was lyophilized to completely remove any remaining water. This is how GPC diagrams were obtained. The peaks of Copolymer–1, Copolymer–2, and B–1 occurred at approximately 25 min with a slight difference. The PLGA–PEG–PLGA triblock copolymer blend has a retention time between that of Copolymer–1 and Copolymer–2 (Figure 4C), and the molecular weight calculated was also confirmed to lay between them.

The independent variables and dependent variable T_gel_ were entered into SPSS 25.0, and an entering multiple linear regression was fitted to them in order to forecast and examine the major and minor influencing factors on T_gel_. Figure 5 shows the results of the regression analysis. Two influencing factors, N_LA_ and M_PLGA_, were omitted due to a lack of association (*p* > 0.5), leaving eight factors M_PEG_, N_GA_, R_LA/GA_, R_PLGA/PEG_, M_n_ NMR, M_w_ GPC, M_n_ GPC and PDI, as potential influencing linear regression factors. Regression ANOVA was performed on the dependent variable t_gel_ with *p* < 0.01, demonstrating that the regression model fits the data set well and that the linear relationship between the explained and explanatory factors in the linear regression model is generally significant.

Linear regression expression: T_gel_ = −0.675 M_PEG_ + 48.61 N_GA_ + 88.555 R_LA/GA_ − 373.281 R_PLGA/PEG_ + 0.136 M_n_ NMR − 0.039 M_w_ GPC + 0.048 M_n_ GPC + 233.856 PDI + 374.517 (R^2^ = 0.987). After undergoing Z–score standardization, the coefficients for each parameter are −14.029 (M_PEG_), 4.478 (N_GA_), 3.916 (R_LA/GA_), −8.063 (R_PLGA/PEG_), 9.506 (M_n_ NMR), −5.104 (M_w_ GPC), 4.828 (M_n_ GPC) and 2.422 (PDI). According to the post standardization analysis, the length of the PEG block plays an important role in the T_gel_ in this fitted linear equation, with a correlation coefficient of an impressive −14.029. It is generally assumed that PEG produces a PLGA–PEG–PLGA hydrophilicity and that the longer the PEG chain, the better the hydrophilicity of the triblock copolymer. However, the fact from statistical analysis is that after PEG reaches around 1500, its hydrophilicity can no longer be increased by increasing the chain length, but instead its hydrophobicity is increased. This is also in good agreement with the nature of PEG in the transition from liquid to semi-solid to solid PEG from low molecular weight (tens) to high molecular weight (hundreds of thousands).

In other words, gelling properties are determined by PLGA and PEG together (R_PLGA/PEG_). The higher the molecular weight of PLGA block or the lower the molecular weight of PEG block, the more hydrophobic the copolymer, which breaks the affinity level equilibrium as the temperature rises and is more susceptible to micelle aggregation and bridging after forming micelles in water. In addition, Chen et al. [68] proved that the PDI size of the PEG block has an effect on its gelling behavior. The larger the PDI, the easier the entanglement of gels at low temperatures, which is also consistent with the regression analysis performed in this work.

### 3.5. Particle Size and XRD

After micronization, the TMZ particle had effectively downsized to 4.4 μm (D_50_) from its original size of 30 μm (D_50_) (Figure 6A), a size decrease of almost 85%. SEM also confirmed that the particle size and appearance of the TMZ crystal changed.

XRD was used to determine if the crystallinity of TMZ had changed following micronization. The results revealed that micro-TMZ and TMZ shared the same peaks, indicating that TMZ and micro-TMZ have identical crystallinity, which was not damaged after the micronization. According to the Debye–Scherrer Formula, the full width at half maximum (FWHM) for various diffraction angles displayed little change, indicating a difference in particle size. It displayed a diffuse X-ray peak of the PLGA–PEG–PLGA hydrogel, indicating that the gel lacked crystallinity [69]. Micro-TMZ remained in a crystalline state in the micro-TMZ@PLGA–PEG–PLGA hydrogel, as evidenced by the fact that its unique diffraction angle persisted during physical mixing of the gel and drug (Figure 6B).

### 3.6. In Vitro Drug Release

TMZ is a super–active alkylating reagent with a highly unstable structure that opens two amide bonds and hydrolyzes into its metabolite MTIC in vitro in a neutral and alkaline aqueous environment gradually [70]; the solubility in pH 7.4 PBS solution is around 2–4 mg/mL [71]. MTIC is a highly active reagent that is only stable at −80 °C for a few days [72]; at body temperature, it promptly transforms into the stable metabolites 4–aminoimidazole–5–carboxamide (AICA) and diazomethane. Diazomethane is a free radical that acts on DNA by alkylating the oxygen atom of guanine at position 6 and the nitrogen atom at position 7 on the DNA molecule, and it is too unstable to be detected [73]. Therefore, the dissolution HPLC detection revealed only two major peaks, TMZ and AICA, with retention times of approximately 15 and 23 min, respectively. Throughout the duration of dissolution, micro-TMZ@PLGA–PEG–PLGA hydrogel continues to dissolve and hydrolyze. TMZ continues to hydrolyze even after samples are collected at each time point prior to detection. We detected the remaining agents of the hydrogel after lyophilization, and the results revealed that neither AICA nor any related chemicals were present. Thus, it is determined that AICA is produced following micro-TMZ@PLGA–PEG–PLGA hydrogel dissolution, and the AICA content should be computed and added back to the TMZ content for the entire dissolution (Equation (7)).
N_AICA_ = N_TMZ_(5)
M_TMZ(AICA)_ = M_AICA_/W_AICA_ × W_TMZ_(6)
Cumulative Release (TMZ) = ∑M_TMZ_ + ∑M_TMZ(AICA)_(7)

Figure 7D illustrates the release characteristics of micro-TMZ@PLGA–PEG–PLGA hydrogels with varying loading levels. After reintroducing the hydrolyzed AICA, the total release of all formulations could reach 100 percent, which is consistent with mass conservation. With increasing drug loading, the duration of drug release increased, from 72 h at 3 mg to 240 h at 250 mg. There was a positive association between hydrogel release length and gel loading TMZ concentration. Gel release curves were fitted to the zero–order release model, the first–order release model, the Higuchi release model, and the Korsmeyer–Peppas release model for five different drug loadings. The results indicated that in the Korsmeyer–Peppas release model, *n* was 2.5723, 1.6442, 1.4379, 1.7075, and 1.7062, which are all greater than 1 in Table 2. Micro-TMZ release in PLGA–PEG–PLGA hydrogels is determined to be concentration–dependent and zero–level release [74]. The calculated regression curves with non-Fickian diffusion and gel dissolution as the rate–limiting step for drug release were statistically significant. The Korsmeyer–Peppas model proved more reliable for hydrogels with 3 and 50 mg/mL drug loadings (R^2^ of 0.8836 and 0.9805, respectively). At low drug loading, TMZ dissolves in the hydrogel in molecular form, and drug release is dominated by diffusion, with gel dissolution constituting the ultimate release characteristic. At high drug loading, TMZ is embedded in the mesh structure of hydrogel in crystalline form, and its release is driven primarily by gel dissolution. After gel dissolution, TMZ is dissolved in the dissolution medium, and the hydrolysis product AICA is also produced.

### 3.7. Thermo-Reversibility and Micelle Reorganization

Figure 8A exhibited heat recovery of PLGA–PEG–PLGA hydrogel by rheometer test. When heated to 50 °C, the modulus of the hydrogel came to the lowest point, indicating the collapse of micelle structure to precipitation. The precipitation could reorganize after being placed at 4 °C for about 25 min, with the continuously climbing up and the re-equal of the loss modulus G″ and the storage modulus G′. G″ went beyond G′ again. The system re-expresses its liquid nature, indicating the reflow of PLGA–PEG–PLGA hydrogel. After five times of heat recovery, the T_gel_ increased from 31.66 to 32.03 °C. This can be explained by the reorganization of the self–assembled PLGA–PEG–PLGA micelles: enhancing confusion to increase the system entropy such that the Gibbs Free Energy decreases and the system becomes thermodynamically stable.
∆G = ∆H − T∆S(8)

When ∆S increases, ∆G drops accordingly, favoring the spontaneity of the reaction.

The heat–precipitated gel can reform to liquid after being cooled back at 4 °C for a sufficiently long time, but the repetitive precipitation–sol transition may cause hydrolysis of the PLGA segment, finally causing the system to be unable to form a gel.

Segment PLGA is a polyester that will hydrolyze into LA and GA under an aqueous circumstance; thus, storing the liquid at a low temperature or freezing it to slow down the hydrolyzation is a good choice before thawing to use. Figure 8B is the freeze–thaw recovery of PLGA–PEG–PLGA hydrogel. T_gel_ and T_precipitation_ were measurable as well with the increase in temperature. The T_gel_ rose from 31.30 to 31.68 °C, which can also be explained by the micelle reorganization. The ice crystallinity might have damaged the micelle structure, forcing the copolymer to adjust to the crystallinity. When thawing, the coil of the copolymer matched randomly and reorganized to homogeneous micelles, and this might be the reason why the T_gel_ rose and stayed to a certain height. It increased from 31.30 to 31.68 °C without deviation after three times of freeze–thaw procedures and maintained 31.68 °C for the next nine times, further demonstrating the thermodynamically steady state. However, it demonstrates that cryopreservation of PLGA–PEG–PLGA hydrogel solution is feasible before placing under ambient temperature to thaw to a solution before preparation to use; the reorganization that raises the T_gel_ is acceptable.

### 3.8. MTT Assay

MTT assay (Figure 9) was used to determine the PLGA–PEG–PLGA triblock copolymer in vitro toxicity. It demonstrates that the maximal concentration of TMZ solution has a considerable anti-proliferation effect on the U87–MG/Luc cell, whereas the blank PLGA–PEG–PLGA hydrogel has no such effect [75]. TMZ group and TMZ&PLGA–PEG–PLGA group had IC_50_ concentrations of 2390 and 1085 μg/mL, respectively. As reported in the literature [76], the maximal solubility of TMZ in an aqueous solution was about 5 mg/mL, which decreased the viability of U87–MG/Luc cells by approximately 20%. Hydrolysis of PLGA–PEG–PLGA, which creates lactic and glycolic acid, may have provided a local acidic environment [77,78] that slowed the hydrolysis of TMZ, resulting in a lower IC_50_, because TMZ is more stable in an acidic environment. In contrast, the PLGA–PEG–PLGA group exhibited no cytotoxicity between 0.1 and 10 mg/mL, showing that this material has exceptional biocompatibility and the potential to become a biodegradable intracranial drug carrier.

### 3.9. Biosafety and Efficacy of Micro-TMZ@PLGA–PEG–PLGA In Vivo

The biosafety of micro-TMZ@PLGA–PEG–PLGA was investigated following its injection into the maximum capacity (8 μL) of the brains of healthy balb/c mice. After standard cranial injection surgery (Figure 10A), it is evident that the hydrogel will retain its shape after 14 days, but the injection site of the NS group has virtually completely healed, as evidenced by pictures of the anatomical structures. Comparing the micro-TMZ@PLGA–PEG–PLGA group to the NS group and the blank group, the micro-TMZ@PLGA–PEG–PLGA group did not demonstrate evident neuron cell death or inflammation (Figure 10B). This trend may have been exacerbated by the presence of TMZ. As long as intracranial injections are already administered, both body weights of the intracranial injection group have the presence of weight loss (Figure 10C). After 7 days, both the NS group and the micro-TMZ@PLGA–PEG–PLGA group began to regain their body weight. This can be attributed to the adaptation of the brain, which builds tolerance to invasive matters. The biosafety of micro-TMZ@PLGA–PEG–PLGA hydrogel is, in conclusion, acceptable. The hydrogel itself was not identified as a foreign antigen that would cause severe and extensive inflammation.

Figure 10D depicts the entire experimental design for suppressing GBM relapses. The suppression of GBM relapse after resection surgery was explored after various therapies. Figure 10E demonstrates that local administration of micro-TMZ@PLGA–PEG–PLGA hydrogel is more efficacious than intraperitoneal TMZ injection in preventing GBM relapse. Chemiluminescence of GBM model mice was significantly diminished after surgery, indicating that the size and extent of GBM tissue were successfully decreased to an acceptable level. However, GBM cells could not be completely removed, and relapse would certainly occur in the following days (Figure 10G). In 60 days, neither the surgery group nor the surgery + PLGA–PEG–PLGA group had any surviving mice; all animals died from GBM relapse, proving that PLGA–PEG–PLGA was only a drug carrier to build a gel in situ and to maintain releasing the drug without efficacy. Despite the fact that the surgery + TMZ group had a 22% (n = 9) GBM–free survival rate, the toxicity should not be overlooked, since two mice died after continuous 6–day intraperitoneal TMZ injection therapy, even before the incident in the surgery group. Surgery + micro-TMZ@PLGA–PEG–PLGA exhibited an effective suppression of recurrence from the IVIS spectrum, with a GBM–free rate of 40% (n = 10) and a statistically significant survival rate compared to other groups (*p* < 0.05). Based on the Kaplan–Meier survival analysis curve, it was certain that no mice died as a result of local administration of TMZ [79]. Possible causes of mice’s inability to eliminate GBM cells include MGMT expression or other genetic factors [80,81,82,83]. These results suggested that the local administration of TMZ in the form of a drug depot to animals was both safe and efficacious.

## 4. Conclusions

TMZ is a recent anti-neoplastic reagent discovered in the late 1990s that has moderate solubility and lipophilicity compared to traditional chemotherapeutic drugs. It has been submitted for BCS Class 1 and Class 3 classification by Merck & Sharp Dohme Ltd. The most prevalent formulations currently on the market are ordinary capsules and ready–to–use injections. Although TMZ is acknowledged as the first–line medicine in GBM post-surgery treatments, its high loading dose (Temodar^®^, MSD Pharmaceutical Co., Ltd., Hangzhou, China, 100 mg/Capsule), severe systemic toxicity and high recurrence rate yet limited targeting cause these formulations to be restricted, resulting in the majority of patients quitting therapy inevitably.

The main conclusions of this paper are as follows. Two PLGA–PEG–PLGA hydrogels with different T_gel_s were blended in different ratios to achieve the desired customization of the T_gel_. The macroscopic morphology of the PLGA–PEG–PLGA hydrogels was characterized for both drug–loaded and non-drug–loaded cases. The relationship between the molecular block properties and the T_gel_ was also statistically analyzed at the microscopic level in combination with GPC, IGD ^13^C–NMR and rheological tests. The dissolution release of the gels was analyzed using a membraneless dissolution method, and the best drug release model fitting was performed to explain it. Comprehensive animal studies were conducted on the safety and efficacy of the formulation for topical use to inhibit GBM recurrence. This work still has some limitations, one of which is the small number of experimental animals and the fact that no clinical trials have been conducted in humans, and the safety and efficacy in humans are still unknown. Overall, this project provides a new approach to relapse inhibition after chemotherapy for GBM, with the aim of achieving complete inhibition of residual cells, improving the quality of survival and prolonging the life span of patients.

## Figures and Tables

**Figure 1 polymers-14-03368-f001:**
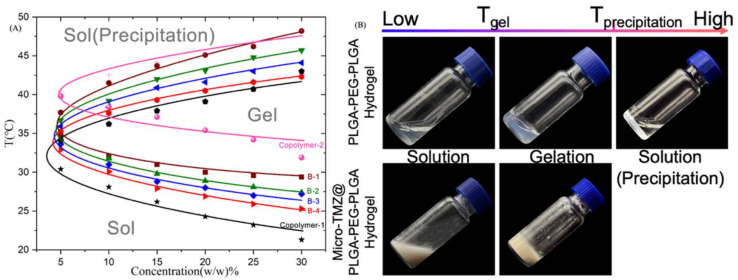
Macro appearance of PLGA–PEG–PLGA thermosensitive hydrogel and preparations. (**A**) Phase diagram of Copolymer–1, Copolymer–2 and blending samples B–1, B–2, B–3, B–4 using Tube–Reverting Method. (**B**) The macro appearance of PLGA–PEG–PLGA hydrogel and micro-TMZ@PLGA–PEG–PLGA hydrogel under 3 temperature stages (below T_gel_, between T_gel_ and T_precipitation_, and beyond T_precipitation_).

**Figure 2 polymers-14-03368-f002:**
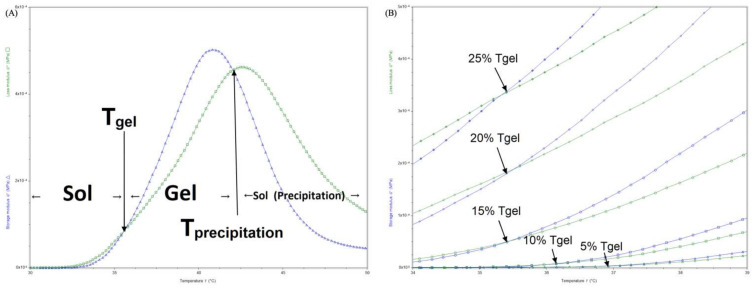
Rheology properties of hydrogel. (**A**) Three phases of PLGA–PEG–PLGA hydrogel in rheology tests. (**B**) Oscillation Temperature Ramp test of Copolymer–2 PLGA–PEG–PLGA hydrogels at concentrations of 5, 10, 15, 20, and 25 wt%. (**C**) Oscillation Temperature Ramp test of Copolymer–1, Copolymer–2 and polymer blending samples B–1, B–2, B–3, B–4. (**D**) Relation between T_gel_s of PLGA–PEG–PLGA hydrogel solution and different blending ratios. (**E**) Oscillation Temperature Ramp test of PLGA–PEG–PLGA hydrogel in loading dosage of 20, 40, 60, 80, and 100 mg micro-TMZ/mL blank PLGA–PEG–PLGA hydrogel.

**Figure 3 polymers-14-03368-f003:**
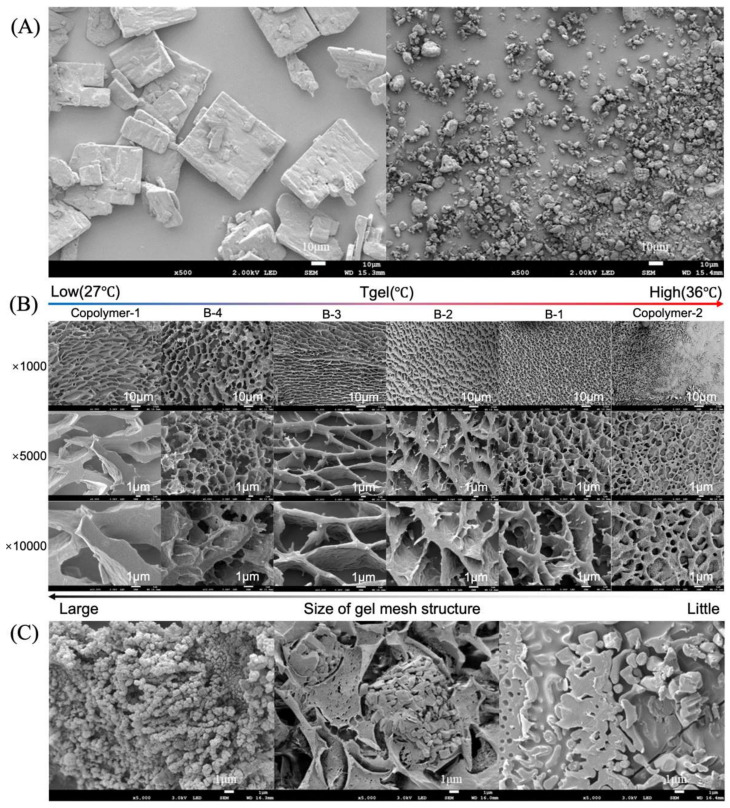
Photographs of SEM and cryo-SEM. (**A**) SEM examination of TMZ crystal micromorphology and micro-TMZ micromorphology. (**B**) Cryo-SEM analysis of 20 wt% PLGA–PEG–PLGA blend samples demonstrates the creation of a distinctive hydrogel mesh structure at 37 °C. (**C**) Three micro-TMZ distribution topologies were identified in the thermosensitive PLGA–PEG–PLGA hydrogel mesh structure.

**Figure 4 polymers-14-03368-f004:**
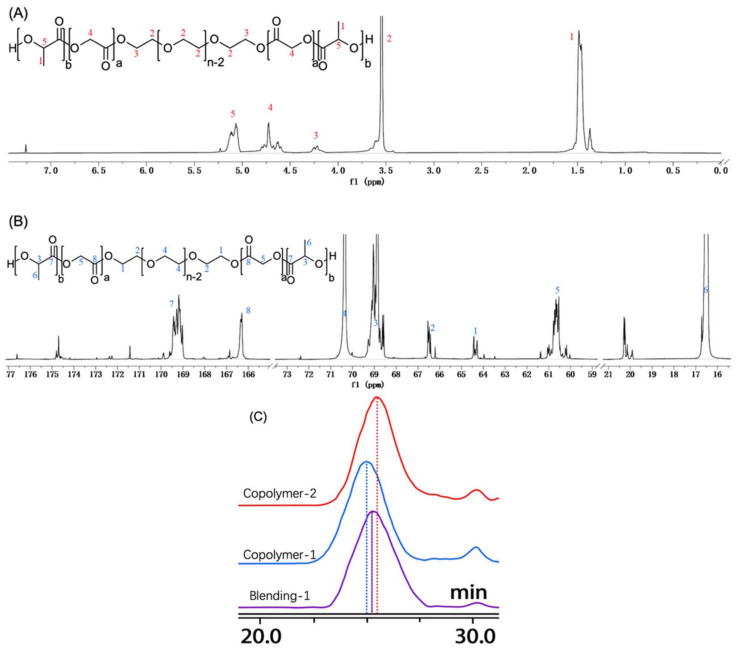
^1^H–NMR (**A**), IGD ^13^C–NMR (**B**) spectrum of PLGA–PEG–PLGA and analysis of each peak integration. GPC diagram (**C**) of Copolymers and B–1.

**Figure 5 polymers-14-03368-f005:**
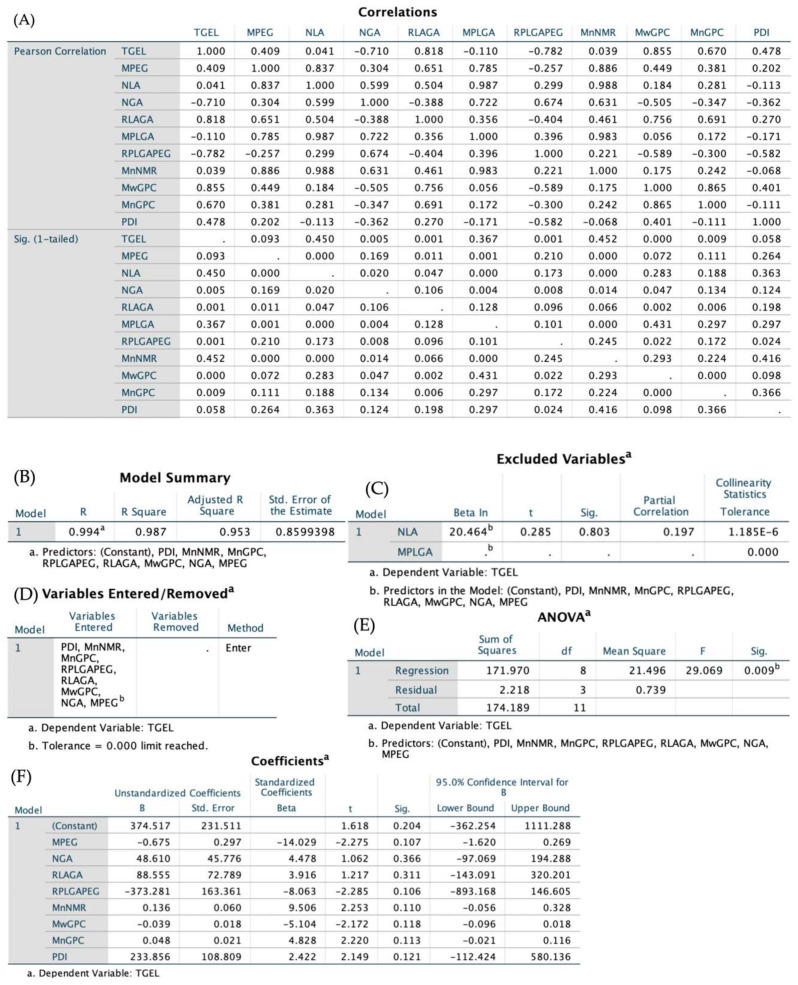
Linear Regression analysis of T_gel_ by M_PEG_, M_PLGA_, N_LA_, N_GA_, R_LA/GA_, R_PLGA/PEG_, M_n_ NMR, M_w_ GPC, M_n_ GPC and PDI. (**A**) Pearson Correlation and significance between variables. (**B**) Goodness of fit of the Linear Regression Model. (**C**) Variables excluded in the linear fitting process. (**D**) Variables that was selected in Enter mode. (**E**) ANOVA test for linear regression. (**F**) Coefficients and the Z−score standardized coefficients of the variables.

**Figure 6 polymers-14-03368-f006:**
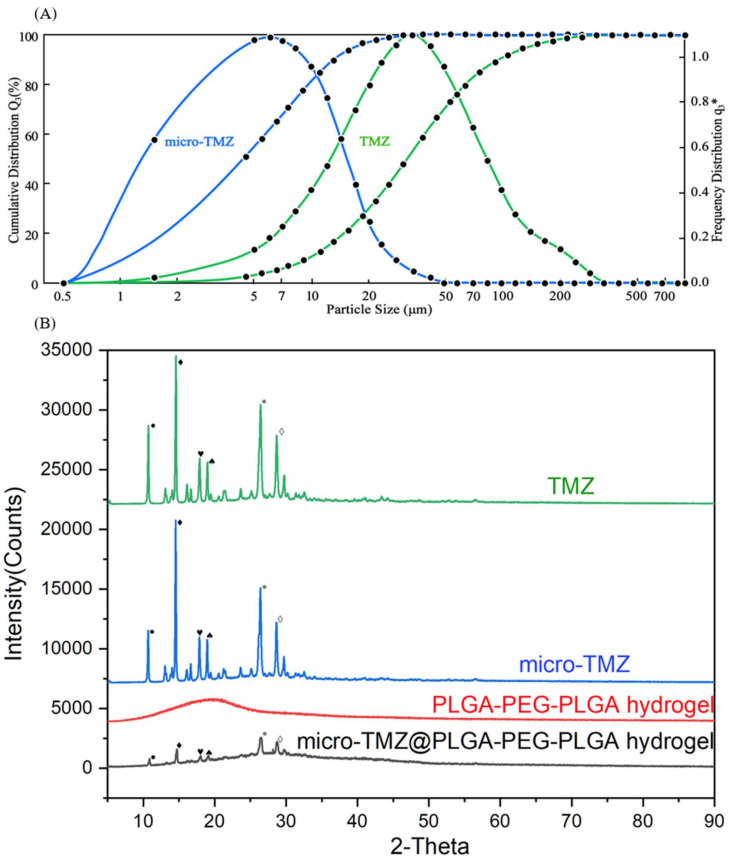
The crystal structure of TMZ and micro-TMZ. (**A**) Measurement of the dry particle size of TMZ and micro-TMZ. (**B**) XRD data for TMZ, micro-TMZ, PLGA–PEG–PLGA hydrogel (blank) and micro-TMZ@PLGA–PEG–PLGA hydrogel (100 mg/mL), the symbols are the unique diffraction angles in each crystallinity.

**Figure 7 polymers-14-03368-f007:**
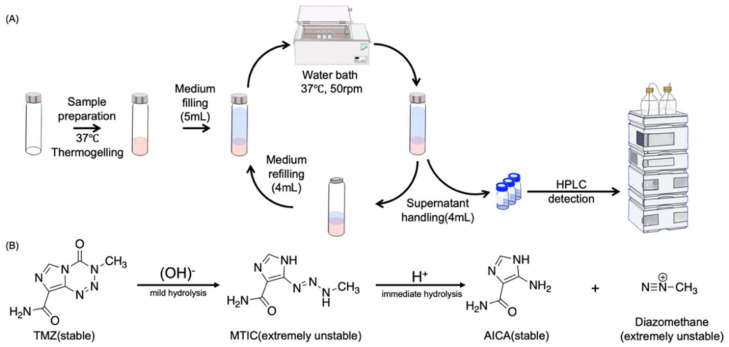
The drug release profile of micro-TMZ@PLGA–PEG–PLGA hydrogel in the pH 7.4 PBS dissolving medium. (**A**) Workflow of the membraneless dissolution process. (**B**) Destiny of TMZ presence under aqueous condition. (**C**) HPLC diagrams of TMZ, AICA, system suitability solution, dissolution contents, and lyophilized contents of micro-TMZ@PLGA–PEG–PLGA from top to bottom. (**D**) Dissolution profile of micro-TMZ@PLGA–PEG–PLGA hydrogel with different drug loadings of 3, 50, 100, 150, 250 mg/mL (n = 3) in PBS (pH 7.4) solution at 37 °C.

**Figure 8 polymers-14-03368-f008:**
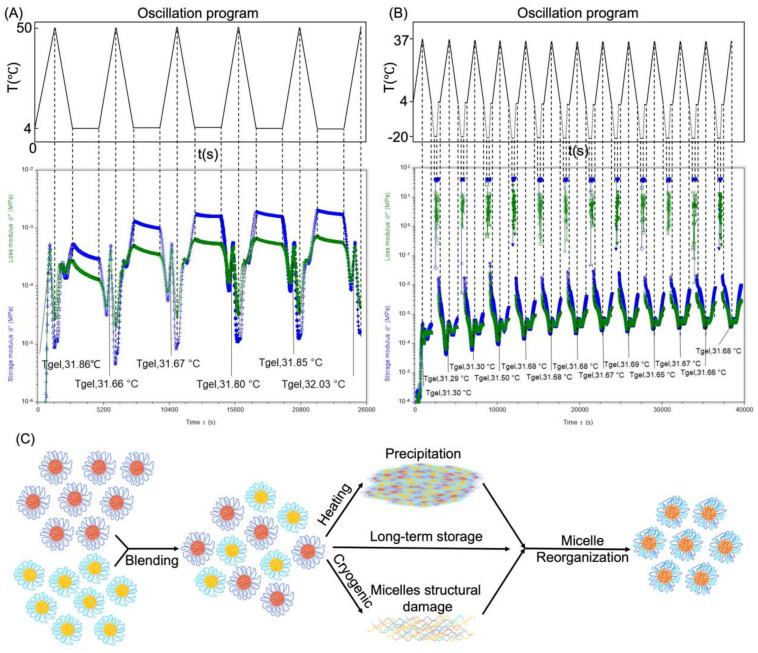
Thermo–reversibility of PLGA–PEG–PLGA hydrogel. (**A**) Heat recovery. (**B**) Freeze–thaw recovery. (**C**) Schematic diagram of micelle reorganization.

**Figure 9 polymers-14-03368-f009:**
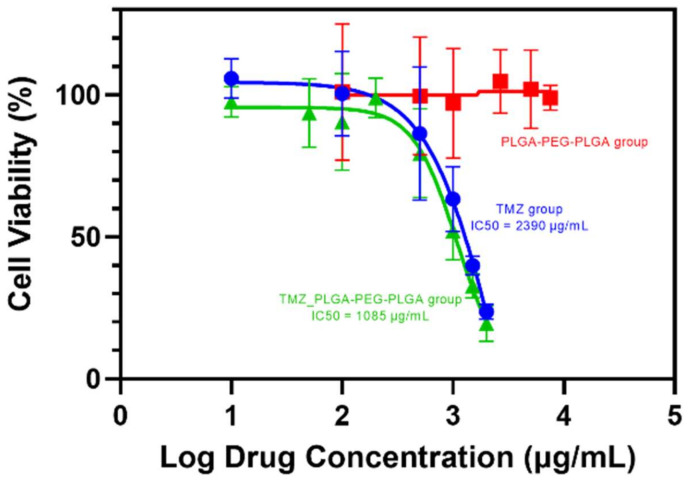
TMZ, TMZ and PLGA–PEG–PLGA, PLGA–PEG–PLGA group anti-proliferation level on U87–MG/Luc cell in vitro as determined by the MTT assay.

**Figure 10 polymers-14-03368-f010:**
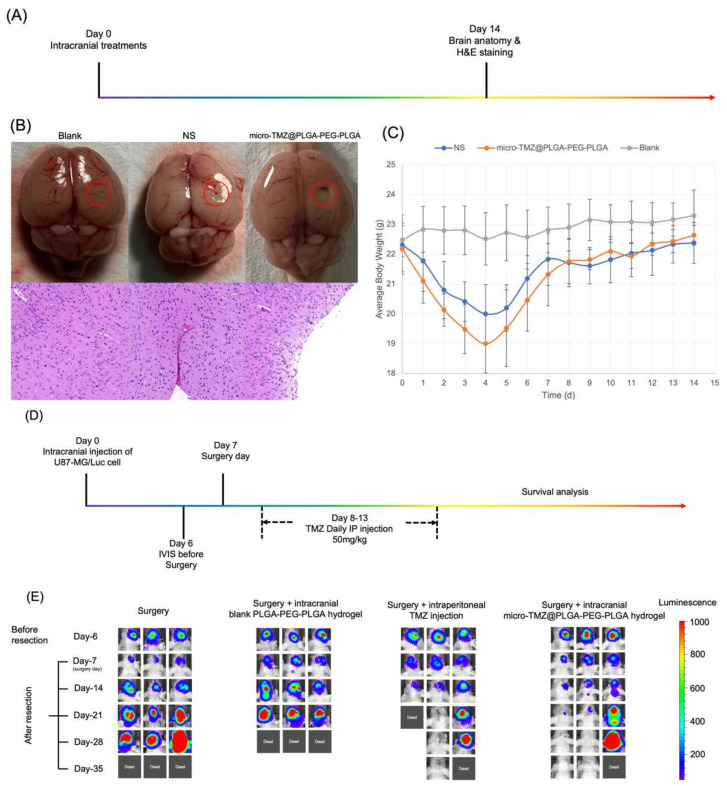
Safety and efficacy experiments of the micro-TMZ@PLGA–PEG–PLGA hydrogel. (**A**) Schedule of biosafety experiment of micro-TMZ@PLGA–PEG–PLGA hydrogel in vivo. (**B**) Macro appearance and H&E staining of healthy balb/c mice brain. (**C**) Bodyweight of balb/c mice under three different treatments (Blank group, NS group, micro-TMZ@PLGA–PEG–PLGA hydrogel group). (**D**) Schematic diagram of the anti-GBM experimental procedure. (**E**) Representative IVIS spectra of U87–MG/Luc cell in different treatment groups, n = 3. (**F**) In vivo image of balb/c nude mice after GBM tumor resection and micro-TMZ@PLGA–PEG–PLGA injection, respectively. (**G**) Survival curve of mice using Kaplan–Meier statistic survival analysis method. (n = 16 for surgery group, n = 10 for surgery + PLGA–PEG–PLGA group, n = 9 for surgery + TMZ intraperitoneal injection group, n = 10 for surgery + micro-TMZ@PLGA–PEG–PLGA group. *: *p* < 0.05).

**Table 1 polymers-14-03368-t001:** Molecular characteristics of series PLGA–PEG–PLGA calculated from ^13^C–NMR and GPC.

Label	M_PEG_	N_LA_	N_GA_	R_LA/GA_	M_PLGA_	R_PLGA/PEG_	M_n_ (NMR)	M_w_ (GPC)	M_n_ (GPC)	PDI	T_gel_ (°C)
Copolymer–2	1616.78	21.84	6.22	3.51	1933.89	2.39	5484.56	8321	7004	1.19	35.61
B–0.5	1389.96	18.56	5.84	3.18	1675.04	2.41	4740.04	7201	6224	1.16	32.65
B–1	1539.34	20.60	6.46	3.18	1857.81	2.41	5254.96	6858	5823	1.18	31.35
B–1.5	1445.62	18.97	6.20	3.06	1724.79	2.39	4895.2	7058	6454	1.09	30.01
B–2	1483.46	19.44	6.41	3.04	1771.53	2.39	5026.52	7240	6162	1.17	29.63
B–2.5	1409.54	18.48	6.12	3.02	1685.52	2.39	4780.58	6997	5917	1.18	28.90
B–3	1553.42	20.38	6.66	3.06	1853.28	2.39	5259.98	7085	6014	1.18	28.70
B–3.5	1565.74	20.57	6.78	3.03	1874.28	2.39	5314.3	6953	5769	1.20	28.60
B–4	1367.3	18.04	6.04	2.99	1648.91	2.41	4665.12	6870	5573	1.23	28.51
Copolymer–1	1424.5	18.28	6.26	2.92	1679.89	2.36	4784.28	6855	5810	1.17	27.25
Copolymer–3	1585.10	19.40	5.68	3.42	1726.24	2.18	5037.58	7579	6143	1.23	37.97
Copolymer–4	1563.54	21.98	6.26	3.51	1946	2.49	5455.54	9465	7869	1.20	I *
Copolymer–5	1480.82	17.79	5.58	3.19	1603.87	2.17	4688.56	8273	6603	1.25	39.14

PS: M_PEG_ and M_PLGA_: molecular weight of PEG block and one PLGA block calculated from IGD ^13^C–NMR. N_LA_ and N_GA_: quantity of LA and GA units in the triblock copolymer calculated from IGD ^13^C–NMR. R_LA/GA_ and R_PLGA/PEG_: ratio of N_LA_/N_GA_ and ratio of 2* × M_PLGA_/M_PEG_. B–0.5, B–1, B–1.5, B–2, B–2.5, B–3, B–3.5, B–4. B–X: blending of Copolymer–1/Copolymer–2 = X (*v*/*v*). I *: water insoluble (under any temperatures, at any concentrations, even dialysis method was tried).

**Table 2 polymers-14-03368-t002:** Kinetic parameters of dissolution profile with different drug loading.

Drug Loading (mg)		3	50	100	150	250
Zero–Order Model	R^2^	0.6202	0.8832	0.9934	0.9889	0.9972
	K_0_ (mg/h)	0.9445	0.7246	0.6122	0.5372	0.3910
	F	8.1644	60.4785	1360.3078	889.7366	4233.0078
	*p*	0.0355	0.0001 *	0.0000 **	0.0000 **	0.0000 **
First–Order Model	R^2^	0.4715	0.7478	0.8096	0.8909	0.8383
	K_1_ (h^−1^)	0.0168	0.0178	0.0184	0.0148	0.0116
	F	4.4616	23.7203	38.2707	81.6265	62.2299
	*p*	0.0884	0.0012 *	0.0002 *	0.0000 **	0.0000 **
Higuchi Model	R^2^	0.8073	0.9569	0.9665	0.9553	0.9696
	K_h_ (h^−1/2^)	10.2892	9.8076	8.4369	7.8597	6.3814
	F	20.9452	177.5321	259.4737	213.4981	382.2087
	*p*	0.0060 *	0.0000 **	0.0000 **	0.0000 **	0.0000 **
Korsmeyer–Peppas Model	R^2^	0.8836	0.9805	0.9922	0.9467	0.9843
	K_p_ (h^−*n*^)	1.8571	3.4940	4.8246	4.0696	4.3371
	n	2.5723	1.6442	1.4379	1.7075	1.7062
	F	37.9687	401.5021	1151.3027	177.6524	754.6880
	*p*	0.0016 *	0.0000 **	0.0000 **	0.0000 **	0.0000 **

α = 0.05. *: *p* < 0.01, ANOVA of regression with significance level; **: *p* < 0.0001, ANOVA of regression with great significance level.

## Data Availability

Not applicable.

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
