# Peer review of "Intracranial In Situ Thermosensitive Hydrogel Delivery of Temozolomide Accomplished by PLGA–PEG–PLGA Triblock Copolymer Blending for GBM Treatment"

_polymers, 2022, doi:10.3390/polym14163368_

Round 1

Reviewer 1 Report

This work addresses important attempts for local delivery of an anti-cancer drug. I will not comment on the biological and most important aspects of the work as I am not at all a specialist. I will thus focus on some particular aspects of the work.

Methodological remarks

Line 512: The authors state that their linear analysis of Table 1 led them to accurate predictions of Tgel.  However, the raw values in Table 1 differ widely in their orders of magnitude with, for example, the Mn(NMR) of order 5000 and the PDI of order 1.   I think that a normalization by replacing each value for a given parameter by its difference with the mean value for this parameter and then dividing this difference by the standard deviation of all values for this parameter would have been a better choice. I have verified that the linear predictor with the raw values in Table 1 is less accurate than with the normalized values. I suggest that the authors test this normalization and decide whether or not it has some importance.

About the different kinetic models of drug release

As a non-specialist, I am surprised that the kinetics of drug release is addressed by so many empirical models devoid apparently of any theoretical basis. (If am wrong about this, then it would be wise of stating rapidly what is their theoretical basis.)  Of course, it cannot be asked to the authors to develop from scratch a more comprehensive analysis in the scope of this work. But is it really the case that nothing has been done in this domain to consider in a unique model (1) the existence of micro-crystals embedded in a gel, (2) the dissolution of these crystals, (3) the diffusion of the molecules in and out of the gel, and (4) the dissolution of the gel itself? Such a model would then explain all situations without considering any ad-hoc empirical equations.

About the thermodynamic argumentation

Lines 614-618:

This can be explained by the reorganization of the self-assembled PLGA-PEG-PLGA micelles: enhancing confusion to increase the system entropy so that the Gibbs Free Energy decreases and the system becomes thermodynamic stable. G=H - TS. When S increases, G drops accordingly.”

Sorry but the thermodynamic argumentation seems poor. First, since G = H – T S, G=H - △(T S)  which is equal to G=H - TS  only when T is constant not when T varies from 50 °C to 4 °C. In addition, figure 7 shows apparently what is this micelle reorganization and what is shown is more an increase than a decrease of order. If my interpretation is wrong, this means that the text is not clear.

Minor  points

Line 115: Express more accurately the meaning of wt %

Line 129: Why “Intriguingly” before “we hypothesized” ?

Line 581: Since the statement is about the increase of the duration of drug release, it should be expressed as: “from 72h at 3 mg to 240h at 250 mg”

Line 594: This sentence seems incomplete: “At high drug loading, TMZ is embedded in the mesh structure of hydrogel in crystalline form, primarily by gel dissolution

Lines 612-613:  The following sentence is awkward: please rephrase it. Also, what are G’ and G’’?

“… with the continuously climbing up and the re-equal of G’ and G’’, G’ went over G’’ again, indicating the reflow of PLGA-PEG-PLGA hydrogel

Line 617: “… becomes thermodynamiCALLY  stable”.

Line 619: “… after being cooled back at 4 °C for a sufficiently long time…”

Note that there are many places where the expression could be improved.

Reviewer 2 Report

This work provides plenty of information on the developed hydrogel, with or without TMZ, as well as interesting biological results. Generally, I propose some improvements:

-English requires improvement in several parts of the text.

-Lines 48-49 do not make sense to me.

-Lines 64-73 are a bit confusing. Line 64, for each segment's Mw before synthesizing, does not make sense. Line 69, till what gets dissolved?

-The intro does not describe how the hydrogel technology can be incorporated into GBM post-surgery treatment. In Line 75, a gelation temperature is suddenly mentioned, followed by Line 78 that TMZ releases slowly from the hydrogel.

-Materials and methods section is too detailed in terms of method development and description and contains discussion. Several parts can be transferred to existing or maybe new result sections, but also probably moved to supplementary information.

-Line 115, did the authors use 450nm filters every time for hydrogel preparation? Did this lead to polymer loss and as a result, the concentration of the polymer in biological assays was lower? This must be mentioned in the relevant discussion parts.

-Line 133, is "between" the right word?

-2.3.6 section, more details on particle size analysis are required. What type of scattering, software etc were used?

-2.4.4 section, in what solvent was TMZ dissolved? Polymer concentration was 1mg/mL in all TMZ concentrations? What is the reason for this?

-2.4.7 section, in what solvent was TMZ dissolved? Is 50mg/kg the standard TMZ scheme for GBM?

-Results should be named Results and Discussion.

-Lines 398-406 are a bit confusing.

-Figure 2, axes are not very clear in most figures.

-Line 617, the 2nd thermodynamic law should be given as an equation.

-3.8, was there any statistically significant differences? Units in Line 646 are incorrect. Also, it does not look to me from the graph that the IC50 values for TMZ and TMZ-in-hydrogel cases are 2.390 and 1.085 mg/mL respectively.

-Figure 8, SDs for several concentration points are too high, especially for the polymer alone.

-Conclusions are too long. I believe that the second paragraph is a description of the final results and should be incorporated into discussion.
